# A Comprehensive Review of Microalgae and Cyanobacteria-Based Biostimulants for Agriculture Uses

**DOI:** 10.3390/plants13020159

**Published:** 2024-01-06

**Authors:** Amer Chabili, Farah Minaoui, Zineb Hakkoum, Mountasser Douma, Abdelilah Meddich, Mohammed Loudiki

**Affiliations:** 1Water, Biodiversity, and Climate Change Laboratory, Department of Biology, Faculty of Sciences Semlalia, Cadi Ayyad University, Bd Prince Moulay Abdellah, Marrakesh 40000, Morocco; chabiliamer@gmail.com (A.C.); minaoui.farah@gmail.com (F.M.); zineb.hakkoum@gmail.com (Z.H.); 2Polydisciplinary Faculty of Khouribga (FPK), Sultan Moulay Slimane University, Khouribga 25000, Morocco; douma_mountasser@yahoo.fr; 3Laboratory of Agro-Food, Biotechnologies, and Valorization of Plant Bioresources, Department of Biology, Faculty of Sciences Semlalia, Cadi Ayyad University, Bd Prince Moulay Abdellah, Marrakesh 40000, Morocco; a.meddich@uca.ma

**Keywords:** microalgae, cyanobacteria, green microalgae, diatoms, biostimulants, sustainable agriculture

## Abstract

Significant progress has been achieved in the use of biostimulants in sustainable agricultural practices. These new products can improve plant growth, nutrient uptake, crop yield and quality, stress adaptation and soil fertility, while reducing agriculture’s environmental footprint. Although it is an emerging market, the biostimulant sector is very promising, hence the increasing attention of the scientific community and agro-industry stakeholders in finding new sources of plant biostimulants. Recently, pro- and eucaryotic microalgae have gained prominence and can be exploited as biostimulants due to their ability to produce high-value-added metabolites. Several works revealed the potential of microalgae- and cyanobacteria-based biostimulants (MCBs) as plant growth promoters and stress alleviators, as well as encouraging results pointing out that their use can address current and future agricultural challenges. In contrast to macroalgae biostimulants, the targeted applications of MBs in agriculture are still in their earlier stages and their commercial implementation is constrained by the lack of research and cost of production. The purpose of this paper is to provide a comprehensive overview on the use of this promising new category of plant biostimulants in agriculture and to highlight the current knowledge on their application prospects. Based on the prevailing state of the art, we aimed to roadmap MCB formulations from microalgae and cyanobacteria strain selection, algal biomass production, extraction techniques and application type to product commercialization and farmer and consumer acceptance. Moreover, we provide examples of successful trials demonstrating the beneficial applications of microalgal biostimulants as well as point out bottlenecks and constraints regarding their successful commercialization and input in sustainable agricultural practices.

## 1. Introduction

Global agricultural production and consumption are anticipated to increase by 60% in 2050 [1], an increase that lines up with the augmentation of necessities, especially food production. Rather than addressing the problems of resource use, current agricultural practices only focus on increasing yields, suggesting the excessive use of chemicals fertilizers [2,3]. The overuse of chemical inputs ultimately alters the quality of soils, diminishing their fertility and weakens microbial activity within [4]. Emerging solutions were suggested to improve crop yields, in particular biotechnological ones as they have the ability to revolutionize agricultural systems and contribute to solving current and future problems [5,6]. The use of such bio-based and renewable products that stimulate plant growth through different mechanisms is already a well-established reality for the cultivation of a variety of agricultural crops [7]. Soil conditioners, organic fertilizers, biofertilizers, and biostimulants were suggested as emerging and ecofriendly solutions. Biostimulants have the potential to naturally promote plant growth, boost soil fertility, and improve microbial activity in the soil [8]. The earliest definitions described plant biostimulants as materials or agents different than fertilizers, that when used at low quantities, stimulate plant growth [9]. Recently, the European Biostimulants Industry Council (EBIC) defined plant biostimulants as substances or microorganisms that, when administered to plants or the rhizosphere, can enhance natural mechanisms for nutrient uptake efficiency, tolerance to abiotic stress, and crop quality regardless of nutrient amount [10]. Biostimulants can be applied at low doses and they can affect plants’ physiological processes through different metabolic pathways, whereas biofertilizers comprised of natural substances and microorganisms can prompt plant growth and affect soil fertility [11,12,13].

Regardless of the amount of nutrients present, biostimulants are effective in small concentrations for crop trait improvement; moreover, biostimulants can be obtained from both organic and inorganic sources [12]. Organic sources include a diverse group of substances, such as protein hydrolysates, amino acids, humic acids, biopolymers, algal extract or living microorganisms as bacteria, yeasts, fungi and microalgae, while the inorganic sources include beneficial chemical elements as trace elements or inorganic salts in the example of phosphite salts or silicon [14,15]. Apart from that, biostimulants are versatile when it comes to application methods, which can range between direct use as inoculum to use as extracts or hydrolysates.

Cyanobacteria and eukaryotic microalgae are not homogeneous monophyletic groups but rather belong to diverse bacterial or eukaryotic evolutionary lines and phylogenetically distinct groups. Due to their diversity and metabolic plasticity, microalgae represent a potently fertile source of high-value-added metabolites, including proteins, amino acids, enzymes, pigments, polyunsaturated fatty acids, polysaccharides, vitamins, antioxidants and phytohormones [13,16]. As a result, MCBs represent one of the promising solutions for their potential as growth promoters, biotic and abiotic stress alleviators [13,17]. To date, research into MCBs is prevailing compared to macroalgae-based ones that are mainly harvested from marine waters and have been well explored. In fact, microalgae represent a promising alternative and viable platform of biostimulants in addition to the option of producing specific bioactive molecules under controlled conditions [18]. In agriculture, farmers’ interests are increased in using biostimulants and biofertilizers [2,13,16], as several MCBs are available in the market and already in use, predominated by marine and freshwater microalgae-based products, while the exploration of the biostimulant potential of soil microalgae is in its early stages.

The purpose of this paper is to offer a comprehensive review on MCB application in the cropping system. With this aim, the process of formulating MCBs is thoroughly road mapped from strain selection, biomass production schemes, extraction and application methods, and mechanisms of action to final product commercialization. Moreover, several examples of pro- and eukaryotic microalgae strains as sources of biostimulants and microalgae products on the market are highlighted. We also illustrate the ways in which cyanobacteria and microalgae can act as biostimulants, as well as some of the persistent limitations to their widespread use in agriculture.

## 2. Microalgae-Based Biostimulants: From Phototrophic Microorganisms to Biostimulants

### 2.1. Selection of Microalgae Strains with High Biostimulant Potential

Despite the important potential inputs of microalgae in agriculture, choosing adequate and promising strains still poses multiple challenges. Selecting highly potent cyanobacteria and microalgae strains is subjected to several criteria. One of the most important criteria is to achieve a fast and homogeneous growth and productivity as well as an easy cultivation scheme (growth in a nutrient medium and waste resources) to produce sufficient biomass for crop trials. Strains with a high cell growth rate are recommended for their relatively short doubling time. Microalgae such as *Chlamydomonas reinhardtii*, *Chlorella vulgaris*, *Dunaliella tertiolecta*, *Haematococcus* spp., and *Scenedesmus* spp. are well known for their high growth rate and short doubling time [19]. Adaptation to a large spectrum of temperature, light, moisture and daily and seasonal variations is also necessary, especially in the case of open production systems under an arid climate. The selection and choice of microalgae strains also depend on the metabolic growth model. For instance, in a photoautotrophic model, strains with high carbon dioxide fixation and high light use efficiency are favored, whereas using conventional and cheap carbon sources is highly recommended in heterotrophic model (a biorefinery approach). Furthermore, selected strains must have remarkable physiological and biochemical traits, such as atmospheric N_2_ fixation in the case of cyanobacteria as it contributes immensely to nitrogen inputs in soils [20]. Additionally, microalgae and cyanobacteria are capable of symbiotic interactions with soil microorganisms which are highly potent as they can co-exist in the phycosphere while benefiting from each other’s production of bioactive compounds (e.g., exopolysaccharides, amino acids, proteins, and vitamins) or high auxin- and/or cytokinin-like activity [21,22]. Overall, strain selection highly influences production modes and technologies, in addition to affecting the production of added-value metabolites and controls the choice of processing methods [23].

### 2.2. Microalgae Cultivation and Biomass Production

Due to their capacity to produce primary and secondary metabolites, microalgae are referred to as microscopic machineries offering many advantages exploitable in many sectors. Microalgae can be cultivated under different metabolic growth models and production systems depending on energy and carbon sources, including autotrophy, heterotrophy, mixotrophy, and photoheterotrophy [24,25]. The cultivation of microalgal biomass can be achieved either by open or closed production systems, with both offering advantages as well as drawbacks. For instance, the use of open production systems includes using natural waters and fabricated aquaculture systems in which construction fees, maintenance costs and energy consumption are relatively low. However, the algal biomass yield is usually lower and the open pond is always subject to a higher risk of microbial contamination due to direct contact with air [26]. On the other hand, closed production systems emphasize using column, tubular, and flat plate photobioreactors, whereas construction fees, maintenance fees and energy consumption are relatively high in comparison to open systems, but the biomass yield is higher [25]. Exorbitant costs of biomass production, use of large volumes of water, added to high energy inputs are three of many constraints making microalgae biomass production and resource use economically inefficient. Moreover, there is an additional cost due to the nutrient supply, notably the nitrogen source, as it was reported that the production of 1.8 tons of biomass/year requires a total of 16,160 EUR/year for the nitrogen source [27].

The need for new approaches to production without extra water and a greater energy footprint is imperative. In this context, approaches such as biorefineries and a circular economy can be integrated to address these challenges. In this case, a microalgae biorefinery can be defined as a process in which the complete utilization of algal biomass is attainable after its production, and this concept is based on the sustainability of microalgal production by integrating the production along with recovering industrially valuable molecules [28].

To achieve the above, the use of treated wastewater to produce microalgae biomass is one of many promising solutions, since using wastewater as a production medium provides a culture with necessary nutrients such as phosphorus and nitrogen, thereby permitting a decrease in the production footprint [21,29]. Table 1 encompasses some successful experiences in growing microalgae using biorefinery approaches while exploiting the produced biomass as biostimulants.

The biorefinery approach can also be incorporated in agriculture, since successful experiences of microalgae cultivation with agricultural effluents have been reported. Nwuche et al. [34] disclosed that *Chlorella sorokiniana* cultivated in membrane-filtrated palm oil mill effluent (POME) produced a higher dry cell weight among tested batches and control. Moreover, when palm oil mill effluent wastewater was used as a culture medium for *Botryoccoccus sudeticus* and *Chlorella vulgaris*, it was found that biomass productivity and lipid yield were significantly increased, especially in the case of *B*. *sudeticus* [35]. Successful trials using wastewater-grown microalgal biomass as biofertilizers demonstrated the efficiency of the approach—for instance, the use of wastewater-grown cyanobacteria and microalgae biomass as wheat biofertilizer resulted in an increase in available nutrients, and microbial biomass carbon in the soil, whereas significant increases in plant and spike dry as well as the values of 1000-grain weight by 7–33%, 10%, and 5.6–8.4%, respectively [36]. Recently, in an attempt to produce biostimulants and biofertilizers with a zero-waste process, Ferreira et al. [37] used wastewater-grown *Tetradesmus obliquus* biomass as biofertilizer in wheat crops, resulting in an increase in the germination index.

Just as the use of treated wastewaters or agricultural effluents for microalgae cultivation offers advantages, it is crucial to check biomass for pathogens, heavy metal concentration as well as contaminants of emerging concern before any use in agricultural practices [38]. Once microalgal biomass is produced with sufficient quantities, it can be directly used or processed via numerous techniques to formulate biostimulants.

### 2.3. Extraction and Application Methods of Microalgae- and Cyanobacteria-Based Biostimulants

As extracts offer many advantages efficiency wise, the use of cyanobacteria and microalgae extracts as biostimulants is gaining its place in agricultural practices. The main advantage of extraction-based methods is the removal of cell walls which assist the release of intracellular bioactive compounds. The extraction of bioactive molecules can be achieved via several methods. The extraction process comprises the penetration of the solvent into the matrix, solute dissolution and release out of the matrix to be collected afterwards [39]. Extraction efficiency depends on numerous factors including solvent properties, material size, extraction temperature and duration [40,41].

Michalak and Chojnacka [42,43] reported the development of numerous extraction methods aiming to harvest biologically active compounds from the algal cellular matrix. The process of extraction is not unidirectional but it goes through three steps: biomass pretreatment, extraction, and formulation of the extract. Microalgal biomass pretreatment consists of two steps, first is washing and/or drying and second is biomass processing for the extraction. The first steps consist of washing the biomass to eliminate any bonding particles and then drying it either via solar drying, freeze-drying or convective drying [44], whereas the second step consists of processing the biomass by disrupting cell walls to release bioactive compounds, hence increasing the extraction yield [45]. Cell wall disruption can be achieved through three main pathways: mechanical/physical, chemical, and enzymatic disruption [23,42].

Extraction methods comprise traditional methods based on extraction with water such as autoclaving, boiling, and homogenization, plus hydrolysis methods including alkaline, neutral, and acid hydrolysis. Conventional solvent extraction includes liquid–liquid extraction, liquid–solid extraction, and Soxhlet extraction. Additionally, novel extraction methods or assisted methods include microwave and ultrasound-assisted extractions, as well as supercritical fluid and pressurized liquid extractions (Figure 1) [16,42].

The comparison of disruption techniques as well as extraction methods revealed an effect of disruption/extraction method choice on yield and quality of the extracts. In a comparison between disruption techniques, Lee et al. [46] compared autoclaving, bead-beating, microwaves, sonication, and a 10% NaCl solution. Results approved the effect of disruption technique choice, whereas lipid content differed significantly among techniques in which using microwaves resulted in the highest extraction yield. Furthermore, the effects of extraction method choice on the biostimulant effects of microalgae extracts are not yet well documented, as available studies indicate that using different extraction processes method wise and solvent wise affect the biostimulant potential of microalgae.

The comparison of extractor systems also revealed an effect of extraction method on biostimulant performance. For example, Navarro-López et al. [47] compared different extractor systems to detect the optimal formulation of a *Scenedesmus almeriensis*-based biostimulant. They revealed that the use of organic green solvents (acetone or ethanol) resulted in a higher germination index in watercress seeds compared to distilled water and ethanol:hexane:water mixture.

The cost/efficiency criterion differs among extraction methods; for example, putting together chemical, mechanical, physical, and enzymatic methods, the cost/efficiency increases, respectively [16]. Novel techniques focus mainly on solvent-free methods or non-toxic solvent use, which makes them environmentally friendly approaches. Nevertheless, investigating the potential to scale-up from laboratory to industry pilot as well as a cost–benefit analysis of MCB extraction are critical steps to take into account before mass production and formulation of such biostimulants.

Microalgae- and cyanobacteria-based biostimulants can be administrated either in the form of extracts, dry biomass, or whole cultures; spent medium or supernatant; cell suspensions [48]. Therefore, the application method depends on the condition of the biostimulants. These forms can be applied via several application methods such as foliar spray eligible for use on foliar surface, seed treatments or primers used on plant seeds, and through fertigation or soil drench by flooding planted soils (Figure 2). Application of MCBs in the way of foliar spray has been shown to enhance plant growth and yield by improving photosynthesis, nutrient uptake, and stress tolerance [49,50,51,52]. Moreover, MCBs can also be used as seed priming or treatments improving seed germination and seedling vigor, leading to better plant growth and yield [51,53,54,55,56]. Extracts, whole cultures, spent/supernatant medium, and cell suspensions can also be used through soil drenching, which improves soil health and fertility, nutrient availability for plants as well as enhancing soil microbial activity, nutrient cycling, and plant growth [57,58,59]. In a similar manner, they can also be used in hydroponic systems to improve nutrient uptake and plant growth in soilless environments [60,61,62].

Hence, the biostimulant effects may vary to a marked extent depending on the extraction and/or application method. Furthermore, species, season, sampling site, environmental conditions, and culture conditions, especially energy, carbon and nitrogen supply, are all variables affecting the content and concentration of active compounds in algae [23,24].

In addition to the cultivation of microalgal biomass, pretreatments and extraction, another often overlooked challenge in the process of formulating MCBs is the storage and shelf life of these products. Stirk et al. [63] demonstrated the influence of storage time and conditions on the bioactivity of freeze-dried *Chlorella vulgaris* biomass. The key findings highlighted by the authors were that the storage time, temperature, and lighting conditions affected root stimulation, antioxidant and antibacterial activity of *C. vulgaris* over storage time, which alluded that those bioactive metabolites are prone to be degraded with long storage periods.

### 2.4. Mechanisms and Modes of Action of Microalgae- and Cyanobacteria-Based Biostimulants

Mechanisms underlying the effects of biostimulants on plants in general remain insufficiently elucidated. As efforts are still undergoing, the complexity of studying such mechanisms is still challenging. The diversity of compounds or their complexity has made tracing mechanisms that underlies the biostimulant effect rather complicated. However, biostimulant mechanisms of action in the case of microorganisms can be categorized into two categories: direct effects and indirect effects. The first effects incorporate the synthesis of bioactive molecules increasing nutrient uptake and stress alleviation, while indirect effects incorporate physiological traits of microorganisms like phosphorus solubilization and nitrogen fixation [64,65].

The modes of action will differ according to the nature of the substance enclosed in the biostimulant product [66]. Bhupenchandra et al. [67] indicated that the possible modes of action can be correlated to several physicochemical modifications in plants, such as decreased membrane lipid peroxidation, increased chlorophyll content, and improved antioxidant activities. Biostimulants can be considered as enablers that can affect plants either directly or indirectly. Direct effects encompass photosynthesis stimulation, upgrading nutrient uptake efficiency, gene and metabolic pathway regulation, and modulating phytohormone excretion, while indirect effects include soil microbiome modulation, soil structure improvement, and organic matter degradation [68,69] (Figure 3).

However, the shortcomings in understanding how biostimulants work can be addressed using new methods such as omics approaches. For instance, computational metabolomics tools have been successfully used to reveal mechanisms underlying the effect of biostimulants on maize plants under drought stress. Results unveiling those alterations in primary and secondary metabolism have led to an enhancement of drought resistance traits which is due to a biostimulant-induced remodeling of the maize metabolism [70].

## 3. Microalgae and Cyanobacteria Strains as Biostimulants in Agriculture

Several microalgae and cyanobacteria strains were tested as possible biostimulants or sources of biostimulant compounds eligible for use in agricultural practices. Earlier uses of algae lineages in agriculture can be traced back to algalization practices for rice paddies fertilization [71]. Algae can be referred to as the first biostimulants used in agriculture as manure as early as the Roman era to increase soil fertility and promote plant growth [72]. However, starting from 1950, extraction-based products made from various algae species began to replace the use of algae biomass [73]. Many microalgae and cyanobacteria strains possess high potential for biostimulant development due to their ability to produce several metabolites as well as their ecological plasticity. In terms of the use of cyanobacteria and microalgae as biostimulants, cyanobacteria and green alga are the most used, while the use of diatoms is still in its early stages.

### 3.1. Cyanobacteria Use as Plant Biostimulants

The use of cyanobacteria offers many advantages as they possess the ability to cause N_2_ fixation and the production of various organic compounds. Several essays showcased the biostimulant effect of cyanobacteria in a variety of crops. For instance, earlier tests of cyanobacterial biofertilizers/biostimulants capacities revealed that the application, in the form of an extract, of a nitrogen-fixing cyanobacteria *Westiellopsis prolifica* boosted cucumber and pumpkin seed germination as well as plants fresh weight, and nitrogen content in roots, shoots, and leaves [74]. The application of *Nostoc muscorum* in the form of fresh biomass, filtrate, or boiled algal extract triggered significant increases in growth parameters and nitrogen content in wheat, sorghum, maize, and lentil plants [75].

In addition to their role in nitrogen fixation and improving the bioavailability of nutrients in the soil, cyanobacteria can also maintain beneficial interactions with microorganisms, making them highly eligible for symbiosis [76]. The interaction between cyanobacteria and diatoms, i.e., diatom–diazotroph associations (DDAs), has been described. In the example of *Hemiaulus hauckii* and *Richelia intracellularis*, the cooperation benefits both parties as cyanobacterial growth and the nitrogen fixation rate increase in response to carbon transfer from the diatom, whereas the diatom benefit from the nitrogen fixed by the cyanobacteria [77]. Symbiotic interaction between cyanobacteria and green microalgae were also recorded. The synergism between *Botryococcus braunii* and *Nostoc muscorum* revealed an increase of 50% of nitrogen fixation when the two strains were co-cultured under N-deficient conditions, as well as the production of new secondary metabolites [78]. The versatility of cyanobacterial symbiotic interactions makes them highly potent in biotechnological application, especially in the case of biostimulants.

The interest in cyanobacteria’s unique traits and metabolic diversity fueled further research for possible application as biostimulants in the cropping system. Several experiments demonstrated the biostimulant effects of cyanobacterial strains in different crops as growth promoters (Table 2).

Cyanobacteria can also be exploited as stress alleviators. The use of cyanobacteria cellular suspensions or extracts revealed their potential for stress adaptation as they contributed to stimulating defense mechanisms and priming against potential sources of stress (Table 3).

### 3.2. Green Microalgae Use as Plant Biostimulants

Green microalgae displayed several biostimulatory effects on crops. In terms of use in agriculture, *Chlorella* genus is by far the most used strain of green microalgae [92]. *Chlorella vulgaris* is the most used strain in the green algae group for its beneficial metabolites production which can be exploited in agriculture [93]. Early experiences of application of fresh biomass or extracts of *Chlorella* genus strains demonstrated positive effects on a variety of crops.

In recent years, the biostimulant effect of green microalgae is gaining momentum in the scientific community. The research for potential green microalgae candidates is still undergoing and results reveal that indeed this group encompass several traits that can be exploited in agriculture. Table 4 present some examples of the use of green microalgae-based biostimulants as growth promoters.

As microalgae-based biostimulants can be used as growth promoters, they can also be exploited as stress alleviators (Table 5). The utilization of green microalgae-based biostimulants as a strategy to address biotic and abiotic stresses offers many advantages and provides solid sustainable solutions.

### 3.3. Diatoms Use as Plant Biostimulants/Biofertilizers

In spite of the fact that the use of diatoms as biotechnological tools is already established, their use as biostimulants is still in its early stages. Diatoms can be considered as ideal cell machinery for producing a variety of molecules in view of their adaptability to harsh environments. By virtue of that, they can be exploited in several industrial applications [105].

Nevertheless, agricultural trials of the effects of diatoms on crops are not well documented yet and the number of available experiences is limited. Available results revealed that indeed diatoms harness the potential to be used as biostimulants, as results indicated that they can be used as growth promoters or as stress alleviators (Table 6).

On the other hand, the scarcity of results concerning the use of diatoms as biostimulants suggests that their potential as agricultural tools is not yet unveiled. Thus, the need to accentuate research involving their uses as biostimulants or in agriculture in general is crucial to fully exploit their characteristics.

### 3.4. The Use of Microalgae- and Cyanobacteria-Based Consortia/Associations as Plant Biostimulant

Algal and microbial consortia are well known for increasing plants’ capacity to absorb water and nutrients as well as being involved in carbon and nitrogen exchange which accord them the ability to foil negative effects of biotic and abiotic stresses [109]. The logic behind the use of consortia is the increased availability of metabolites produced by the consortia’s components, since the multitude of strains engender more metabolites in comparison with individual strains [110,111].

The biostimulant effects of microalgae and cyanobacteria are well established through several studies and experiments, yet questions about the possibility of enhancing and/or optimizing these effects via associations still surface. Moreover, in soil and aquatic ecosystems, microalgae coexist with several microorganisms such as bacteria. Due to their capacity to produce metabolites such as polysaccharides and phytohormones, microalgae, cyanobacteria and plant-promoting bacteria possess the ability to boost plant growth either individually or in combination, which suggests the possibility of co-culturing or combining microalgae with microorganisms, bringing into being the concept of Microalgae Growth-Promoting Microorganisms (MGPMs). Palacios et al. [112] introduced the concept of Microalgae Growth-Promoting Bacteria (MGPB), which designates the use of bacteria as growth promoters in microalgae culture. MGPB promote microalgae growth through several mechanisms [112,113], such as compensating carbon dioxide, which reduces microalgae enzymatic activity required for carbon dioxide concentration [114]; growth stimulation by production of phytohormone-related compounds in which phytohormone precursor producers such as *Scenedesmus* sp. can induce IAA bacterial production in co-culture [115]; environmental stress mitigation by the production of co-factors following the example of riboflavin production by *Azospirillum brasilense,* which improved the growth of *Chlorella sorokiniana* [116]; and nutrient supply improvement through nitrogen fixation and siderophores production, where it was reported that *Bacillus pumilus*-produced ammonium through nitrogen fixation increased *Chlorella vulgaris* growth [117]. Table 7 regroups some examples of the use of microalgae consortia and associations.

However, the use of microalgae and cyanobacteria consortia or associations calls for further studies as the co-inoculation of microalgal strains or with microorganisms, the use of combined extracts, and application with substances need to be confirmed by determining whether there are synergetic effects between the components of the mixture or on the opposite antagonistic effects.

Additionally, it is crucial to determine each component effect on plant in order to fully understand mechanisms by which these associations stimulate growth or stress adaptation which can also open the door to further optimization of the mixture. Testing these formulations under different pedoclimatic conditions is also crucial to validate their synergetic effects and to decide on the optimal conditions in which they can be applied. Additionally, the adoption of the MGPB approach is still faced with a multitude of challenges to overcome as it still needs to be scaled up from the laboratory scale to the industrial scale, there needs to be control of other bacterial contamination, and its economic and environmental feasibility need to be studied [113].

## 4. Economic and Research Trends in Microalgae- and Cyanobacteria-Based Biostimulants for Agriculture Uses

The research into MCB potential uses across the globe increased significantly in recent years. Health, energy, and human nutrition were the three major applications of microalgae- and cyanobacteria-based products [123]. In terms of agricultural uses, future market insights expect the demand for microalgae in the sector of fertilizers to align with the increase in bioproducts use, whereas the increase in demand is expected to be a 8.7% compound annual growth rate during the period of 2021–2030 [124]. This augmentation, according to future market insights, is mainly due to the increasing interest in MCB usage in agriculture, as agrochemical firms started exploiting microalgae in the formulation of several biofertilizers. The leading market for microalgae in the fertilizers sector across the globe is the U.S. market. The runner-up is Brazil, where organic food consumption is impelling the demand for microalgae in biofertilizers formulations. Meanwhile, in Europe, Germany and Russia are expected to be the highest grossing markets by 2031.

The development of research for sustainable microalgae-derived products is accountable for the pronounced increase in demand for microalgae in the sector of agriculture as well as the increase in microalgae-related patent applications. Earlier increases in patent applications between 2008 and 2013 were ascribed to microalgae-based biofuels, while agriculture-related applications increased from approximately 2015 [125]. According to the same authors, microalgae-related patents were found in different areas of expertise within the agricultural sector, but the largest number of registered patents were in the plant growth area.

Research wise, a bibliometric analysis of the Scopus database indicated that the number of research articles with “microalgae” and “biostimulants” as keywords increased throughout the period of 2016–2023, and the number of articles arose from less than five in 2016 to thirty-tree in 2022 whereas Italy, Spain, Brazil, India, Portugal, and Morocco are leading in terms of the number of documents published per country. By subject area, 35.7% of documents were published in Agriculture and Biological Sciences, followed by Biochemistry with 13.6%, and Environmental Sciences with 10.4%.

## 5. Commercialized Microalgae- and Cyanobacteria-Based Biostimulant Products and Consumer Acceptance

The commercialization of MCBs is already established in the economic landscape. The use of macroalgae is significantly superior to microalgae, since most products are based on seaweeds and marine algae; however, microalgae are not totally absent as some new firms started exploiting them for biostimulant formulations. Several products are being sold and proclaimed as solutions using less water, chemical fertilizers, and pesticides. These products are marketed as plant growth promoters, nutrient uptake efficiency enhancers, and fruit quality improvers. Table 8 summarizes some of the commercially available MCBs and their characteristics.

Recently, a survey about consumer attitude toward microalgae-based agricultural products highlighted that farmers in the region of Almeria (Spain) and Livorno (Italy) were not in favor of building microalgal production facilities in their properties; however, the use of microalgae-based agricultural products such as biostimulants, biofertilizers and aquafeed in both locations was acceptable, especially in Livorno (Italy). Authors also highlighted the influence of farmers knowledge about microalgal biotechnologies and their benefits, whereas acceptance was fairly more positive when farmers gained more information [126].

Another survey conducted by Ruiz-Nieto et al. [127] about farmers’ knowledge and acceptance of microalgae in horticulture greenhouses in the city of Almeria revealed that despite the lack of a solid understanding of microalgae beneficial uses in agriculture, farmers conceived microalgae as having potential benefits to agriculture. In light of this result, it is essential to encourage local efforts to promote the use of microalgae in the cropping system, especially in countries where agriculture is an economic pillar.

Nevertheless, the need to shift towards sustainable tool utilization in agriculture to face climate change is also a driving force that prompts inward movement towards the use of microalgae-based biostimulants.

## 6. Regulations and Legal Framework of Biostimulants for Agriculture Uses

Biostimulants are defined and regulated in the USA and in Europe while they are subjected to international laws in other countries. In Europe, biostimulants are defined as “*a product stimulating plant nutrition processes independently of the product’s nutrient content with the sole aim of improving one or more of the following characteristics of the plant or the plant rhizosphere: nutrient use efficiency; tolerance to abiotic stress; quality traits; availability of confined nutrients in soil or rhizosphere*” [10]. The regulation also defines the nature of components eligible to use as a biostimulant source. Two main categories were defined as materials to produce biostimulants: “Plants, plant parts or plant extracts” and “Micro-organisms”.

Regulations regrouped these categories under fourteen subcategories named Component Material Categories (CMC). Whereas in the USA, regulations define biostimulants as “*substance or micro-organism that, when applied to seeds, plants, or the rhizosphere, stimulates natural processes to enhance or benefit nutrient uptake, nutrient efficiency, tolerance to abiotic stress, crop quality or yield*” [128].

Placing a plant biostimulant product in the European and other markets undergoes a series of requirements to fulfill before commercialization. According to Traon et al. [129], the placement of plant biostimulant products in the EU and third-world countries requires two processes: a registration process and data requirements in which requirements such as characterization and identification, manufacturing process, toxicity data, ecotoxicity data, and environmental fate data are necessary of any authorization for use, although requirements differ depending on the country.

Another crucial step for using biostimulant products in agriculture is the fulfillment of the efficacy criterion which requires demonstrating the efficacy of the biostimulant under determined conditions including crops used, dosage, and optimal conditions of use comprising timing, preferred crop stage, and agro-climatic limitations of efficacy [129]. Recently, a meta-analysis of biostimulant yield effectiveness in field trials reported that the biostimulant category, application method, crop species, climate conditions, and soil properties are all factors affecting the efficiency of biostimulants in field agricultural applications. The authors revealed that among all biostimulant categories, the yield benefit is an average of 17.9%, with soil treatment being the optimal application method. The highest impact on crops yield was observed in vegetable cultivation and in arid climates, while the highest efficiency of biostimulants was recorded in soils with low organic matter content, non-neutral, saline, nutrient-insufficient, and sandy soils [130].

## 7. Bottlenecks and Use Limitations of Microalgae and Cyanobacteria Biostimulants

The interest in MCBs has been increasing in the scientific and economic landscape including among agrochemical firms and crop producers. The encouraging effects of such biostimulants surged the interest to further study their effects as well as implanting them as products in the market. However, several challenges, bottlenecks and use limitations impeding their normalized use, concerns regarding their cost of production, extraction, bioformulation, compatibility and product stability and environmental safety are all obstacles to fully harnessing their potential.

The costs of biomass production are crucial in any form of product formulation as they need to be economically low with available platforms assuring the feasibility of the production and the ease of downstream processing of biomass. Moreover, the ecological footprint and the efficiency of resource use still pose challenges in the face of mass production, especially in the case of expensive nutrients like nitrogen. The use of MCBs in the form of extracts is faced with the need for extraction process optimization energy wise and economically as well as rationalized use of solvents in the case of aqueous extracts. The bioformulation of MCBs is still required to undergo a number of field trials to investigate interactions with soil biological components and environmental safety to assess any possible toxicity and to check for compatibility and stability of the products.

Throughout our review, we indicated the main benefits and drawbacks of using MCBs in agriculture. Accordingly, to address the current challenges these biostimulants need to overcome to be fully operational tools, we suggest the following research and development prospects:

Fully explore the biodiversity of particularly microalgae in aquatic and soil ecosystems, plus broadening tests to include all groups of microalgae in particular diatoms;To address the production challenge, it is necessary to focus on minimizing the costs of production and application through the optimization of production schemes mainly targeting the use of waste resources by adopting biorefinery approach, as well as scaling up application methods to be used on an agricultural scale;In order to fully use the biostimulant potential of microalgae, it is necessary to understand the physiological and molecular mechanisms by which these compounds affect the plant and soil ecosystem by adopting new approaches mainly molecular and omics;The need to fully explore the conditions under which the benefits of biostimulants formulations are optimal, encompassing pedology and agroclimatic conditions, crop type and growth stage, timing of use and frequency of use, as well as revealing the possible interactions with soil components such as bacteria, fungi, and microfauna;For a successful input in agriculture, it is necessary to raise the awareness of farmers regarding the benefits of using these tools.

In addition to these limitations, regulatory issues seem to be another point to address as the lack of standardization, regulatory compliance, and inconsistent definitions and regulations of biostimulants across the globe still hinder the actual implementation of biostimulants in the market.

## 8. Conclusions

Microalgae, both in aquatic and terrestrial ecosystems, put forward numerous exploitation opportunities for their biochemical characteristics and metabolic diversity. Due to the momentum in terms of academic research and successful laboratory essays and field trials, MCBs can now be considered as new solutions and promising alternatives for sustainable agriculture. The wide diversity of microalgae extends their potential as inputs in the cropping system, and provides new chances for discovering new bioactive molecules with valuable biostimulant potential.

Nevertheless, the necessity to explore mechanisms underlying the functioning of biostimulant is critical for further use and development. Although the full potential of these biostimulants is still not completely achieved due to technological limitations, emerging approaches such as green extraction techniques, biorefineries, and the use of industrial byproducts could be exploited to overcome the limitations. Furthermore, the use of these biostimulants in agriculture can indirectly help improve agricultural ecosystem services by affecting the soil and microorganisms’ components.

Thus, the use of MCBs can be deemed as a step for future progress in terms of developing new bio-based technologies that can be used as alternatives and complementary products to improve crop yield and quality.

## Figures and Tables

**Figure 1 plants-13-00159-f001:**
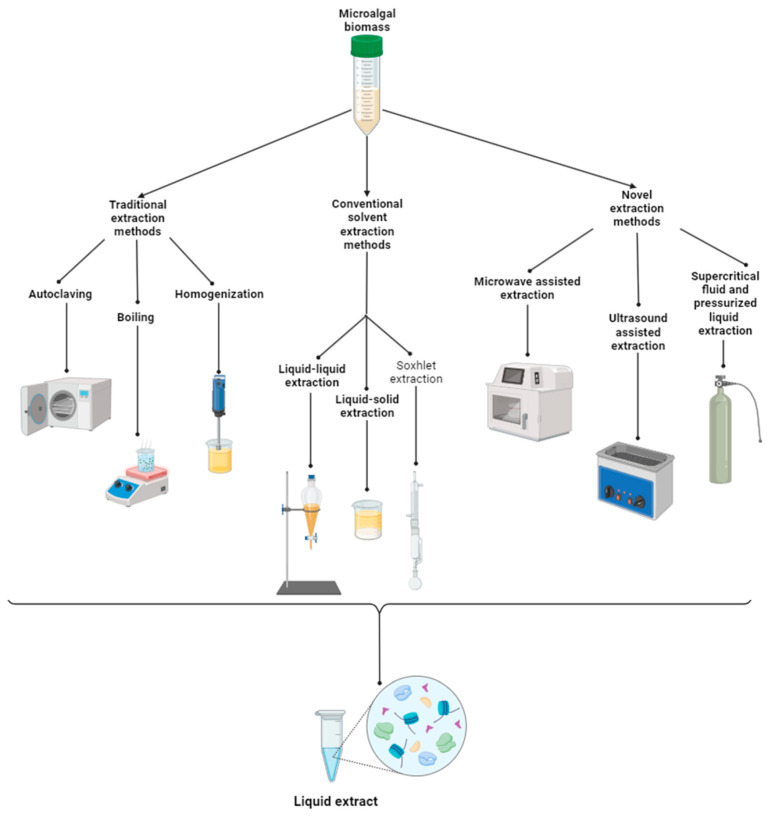
Microalgal biomass extraction methods [42,43] (made with Biorender.com).

**Figure 2 plants-13-00159-f002:**
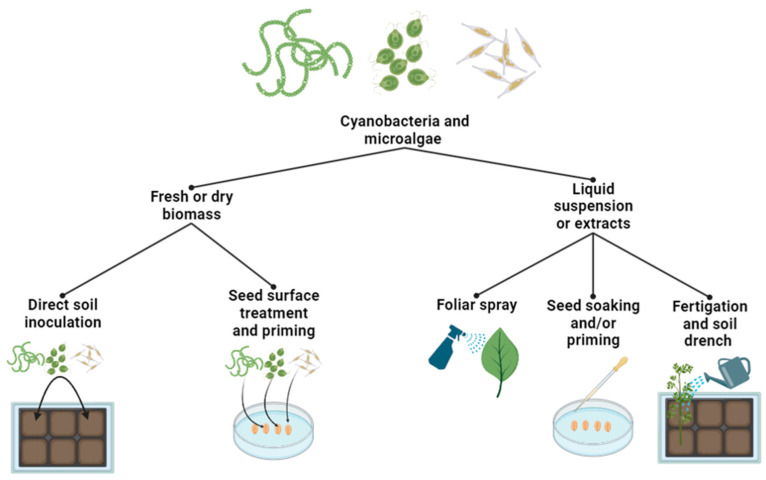
Microalgae- and cyanobacteria-based biostimulant application methods (made with Biorender.com).

**Figure 3 plants-13-00159-f003:**
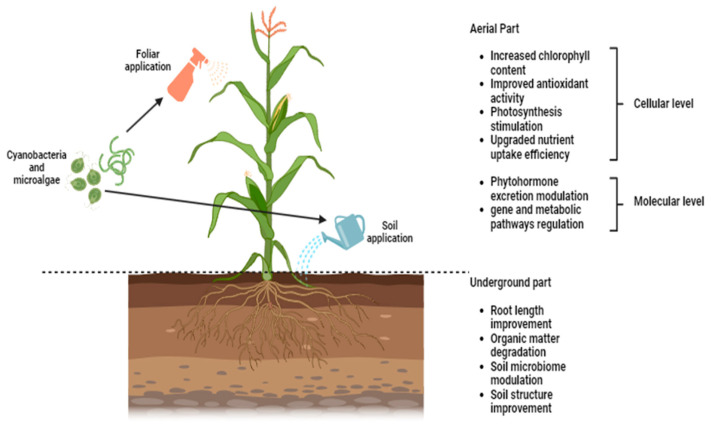
Modes of action of microalgae- and cyanobacteria-based biostimulants on crops (made with Biorender.com).

**Table 1 plants-13-00159-t001:** Biostimulant effects of some green microalgae strains produced via biorefinery approaches.

Green Microalgae Strains	Biomass Growth Medium	Application Methods/Crop Plants or Seeds	Biostimulant Effects	Reference
*Scenedesmus obliquus*	Pretreated brewery wastewater	Centrifuged, ultrasonicated, and enzyme hydrolyzed biomass applied on watercress seeds (*Lepidium sativum*), mung bean (*Vigna radiata*), and cucumber (*Cucumis sativus* L.)	40% higher watercress seed germination indices, suggesting Gibberellin-like effects; 60% higher root formation in mung bean and cucumber, suggesting auxin-like effects; 87.5% higher cotyledon expansion in cucumber, suggesting cytokinin-like effects.	[30]
*Tetradesmus obliquus*; *Chlorella protothecoides*	Piggery wastewater	Fresh microalgal biomass applied on cucumber (*Cucumis sativus*), barley (*Hordeum vulgare*), wheat (*Triticum aestivum*), soybean (*Glycine max*), watercress (*Nasturium officinale*), and tomato (*Lycopersicon esculentum*)	*T*. *obliquus* increased the germination index in barley, watercress, and cucumber, whereas GI increased by 100%; Soybean plant shoot length was increased by 90% when using *C*. *protothecoides*; A slight increase in chlorophyll a content in cucumber and tomato when using *T*. *obliquus.*	[31]
*Chlorella vulgaris* UAL-1; *Chlorella* sp. UAL-2; *Chlorella vulgaris* UAL-3; *Chlamydopodium fusiforme* UAL-4	Secondary-treated urban wastewater supplemented with concentrate	Aqueous extracts applied on watercress (*Lepidium sativum* L.), soybean (*Glycine max* L.), cucumber (*Cucumis sativus* L.), and wheat (*Triticum aestivum* L.) seeds	Increased watercress germination index by 3.5% when using *C*. *vulgaris* UAL-1; Adventitious root formation in soybean seeds promoted by 220% and 493% when using 2 concentrations of *C*. *vulgaris* UAL-1; *C*. *vulgaris* UAL-1 promoted chlorophyll retention in wheat leaves.	[32]
*Chlorella vulgaris* (13–1); *Scenedesmus obliquus* (B2-2)	Untreated municipal wastewater	Algal biomass (intact and broken cells) and culture supernatant applied on tomato (*Solanum lycopersicum*) and barely (*Hordeum vulgare*) seeds	0.5- and 0.25-days faster germination time in tomato and barley seed when using *C. vulgaris*; Faster mean germination time with *S. obliquus* intact cells and supernatant; A higher germination index with broken cell treatment up to 9% higher.	[33]

**Table 2 plants-13-00159-t002:** Cyanobacteria-based biostimulants as growth promoters in different crops.

Cyanobacteria Strain	Application Method	Crop Plant/Seeds	Biostimulant Effects	Reference
*Nostoc* sp.	Soil inoculation with fresh cyanobacteria biomass	Maize (*Zea mays*)	Increased value of coarse aggregates, thus increasing soil stability; Improvement of growth and nitrogen uptake in maize plants; Enhancement of plants’ dry matter yield by 49%; Increased nitrogen uptake and concentration in maize plants tissues.	[79]
*Chroococcidiopsis* SM-04; *Synechocystis* SM-10 *Phormidium* SM-14; *Leptolyngbya* SM-13	Inoculation with fresh cyanobacterial cultures in a hydroponic growth system	Wheat seeds (*Triticum aestivum* var Uqab-2000)	Increased shoot height by 53% with *Phormidium* SM-14 and 42% with *Synechocystis* SM-10 inoculation; Increased root length by 7.2% with *Leptolyngbya* SM-13; Higher root number correlated with higher auxin production.	[80]
*Anabaena vaginicola* ISC90; *Nostoc calcicola* ISC89	Spray with 1% cyanobacterial extracts on soil surface	Squash (*Cucurbita maxima*); Cucumber (*Cucumis sativus* L.); Tomato (*Solanum lycopersicum* L.)	Increased root length with both cyanobacterial extracts in all three plants; Higher plant height in all treatments in comparison with untreated plants; Overall higher root fresh and dry weight in all treatments.	[81]
*Anabaena vaginicola* ISB42; *Cylindrospermum michailovskoense* ISB45; *Trichormus ellipsosporus* ISB44	Foliar spray with 1% cyanobacterial extracts	Peppermint (*Mentha piperita*)	Pronounced higher fresh and dry weight in plants treated with *A. vaginicola* ISB42 and *C. michailovskoense* ISB45; Increased leaf number, leaf area, number of ramifications, and number of nodes with *A. vaginicola* ISB42 and *C. michailovskoense* ISB45 treatments; Enhanced essential oils content in plants treated with *A. vaginicola* ISB42 and *C. michailovskoense* ISB45.	[82]
*Spirulina platensis*	Foliar application with *S*. *platensis*-based commercial product	Eggplants (*Solanum melongena*)	Increased number of flower buds and higher number of fruits per plant; Improved fruit lightness at three days after storage; Enhanced pulp firmness when using low concentrations.	[83]
*Spirulina platensis*	Foliar spray with polysaccharides extract	Pepper (*Capsicum annuum* var. andalus); Tomato (*Solanum lycopersicum* L. Var. metro)	Increased plant size by 20 to 30% in both plants; Improved shoot and root dry weight by 230% in tomato plants against 67% in pepper plants; 57% and 100% respective increases in leaf foliar area in pepper and tomato; Augmented leaves number by 33% and 55% in pepper and tomato, respectively.	[84]
*Microcystis aeruginosa* MKR 0105; *Anabaena sp.* PCC 7120	Foliar spray with intact cyanobacterial monoculture	Willow plants (*Salix viminalis* L.)	Length of shoots increased by 85.8% when using 0.5 g; Improved chlorophyll content, net photosynthesis intensity, transpiration, and stomatal conductance; Enhanced N, P, and K content as well as enzyme activities mainly dehydrogenases, RNase, acid or alkaline phosphatase and nitrate reductase.	[85]
*Arthrospira platensis*	Foliar applications with aqueous suspensions	Lettuce (*Lactuca sativa*)	Enhanced leaf area, fresh and dry weight of leaves with hydrolyzed biomass; Significant Increases in root fresh and dry weight with hydrolyzed biomass in comparison with lyophilized biomass; Overall higher leaf number, lettuce fresh and dry weight with hydrolyzed biomass.	[86]
*Nostoc *sp.; *Tolypothrix* sp.; *Leptolyngbya* sp.	Foliar spray with cyanobacterial hydrolysates in a hydroponic growth system	Basil (*Ocimum basilicum* L.)	34.4%, 31.8%, and 28.7% increases in plants fresh weight with *Leptolyngbya* sp., *Tolypothrix* sp., and *Nostoc* sp., respectively; Treatment with *Nostoc* sp. enhanced root fresh weight by 53%; All treatment increased the number of leaves by 24% and supplemented plants with one more node than the control; Treatment with *Tolypothrix* sp. improved plants height with an increase of 20%.	[62]

**Table 3 plants-13-00159-t003:** Cyanobacteria-based biostimulants as stress alleviators in different crops.

Cyanobacteria Strain	Stress	Crop Plant	Application Method	Biostimulant Effects	Reference
*Spirulina maxima*	Salinity stress	Wheat grains (*Triticum aestivum* L. cv. Giza 94)	Irrigation with *S. maxima* aqueous extract	Higher antioxidant compound accumulation in stressed grains; Increased tocopherols, total carotenoids, and phenolic compounds; Improved scavenging activity concerning DPPH and ABTS+ radicals in grains processed with the extract.	[87]
*Aphanothece* sp. BEA O935B; *Arthrospira maxima* MSS001	Salinity stress	Tomato (*Solanum lycopersicum*)	Extract formulations applied to plants by soil drench	Treatment with extract formulation at 5% enhanced chlorophyll content in plants subject to lower salinity concentration; Proline accumulation under normal and high saline conditions significantly increased with 5% treatment (+140.5% and 87.89%); Enhanced ROS scavenging enzyme activity notable SOD and CAT with treatment at 5%.	[88]
*Roholtiella* sp. QUC- CCM97	Salinity stress	Bell pepper (*Capsicum annuum* L.)	Foliar application with aqueous extract	Increments in total chlorophyll content in stressed plants compared to the control (1.95–3.35 mg/gFW); Increases in proline levels under all salinity conditions ranging from 5.18 to 38.20%; Enhanced antioxidant capacity under high-salinity conditions; Significant increase in Catalase activity under all salinity concentrations.	[89]
*Oculatella* *lusitanica* LEGE 161147	Salinity stress	Lettuce (*Lactuca sativa*)	Plants grown in a mixture of vermiculite and perlite supplemented at the top with perlite containing *O. lusitanica*	Proline content decreased in stressed plants inoculated with *O. lusitanica*; Increased GSH levels in leaves under salinity and cyanobacteria inoculation; Lower MDA levels in stressed plants grown in the presence of *O. lusitanica;* GDH, GS, and NR activities remained unchanged under saline conditions.	[90]
*Arthrospira platensis*	Drought stress	Cotton plants (*Gossypium barbadense* L. cv. Giza 94)	Foliar application with cyanobacterial extract	Higher contents of chlorophyll a and b as well as carotenoid content; Decreased H_2_O_2_ content in stressed plants by 34% under moderate-deficit irrigation and by 22% under severe-deficit irrigation; Reduced MDA level by 33% and 22% under moderate- and severe-deficit conditions; Increased SOD activity in moderate- and severe-deficit conditions by 17% and 15%.	[91]

**Table 4 plants-13-00159-t004:** Green microalgae-based biostimulants as growth promoters in different crops.

Green Microalgae Strain	Application Method	Crop Plant/Seeds	Biostimulant Effects	Reference
*Chlorella vulgaris*	Fresh and dry biomass mixed with soil	Lettuce (*Lactuca sativa*)	Increased fresh and dry weight; Increased shoot and root length; Improvement of pigments content; Enhanced amino acids and proteins content in seeds.	[94]
*Chlorella vulgaris*	Irrigation with freeze dried biomass solutions	Broccoli (*Brassica oleracea*)	Higher antioxidant activity in 7-day-old sprouts; Increased β-carotene content; Ascorbic acid concentration increased by 37% after 7 days and by 100% after 14 days; Increased sulforaphane concentration.	[95]
*Acutodesmus dimorphus*	Cellular extracts and dry biomass applied as seed primer and foliar spray	Tomato (*Solanum lycopersicum*)	Cellular extracts used as primer improved seed germination; Cellular extracts applied as foliar spray increased plant height and flower number; Dry biomass application improved floral formation and branch number.	[96]
*Scenedesmus quadricauda*	Irrigation with Hoagland solution containing *S*. *quadricauda* extract	Lettuce (*Lactuca sativa*)	Improvement of shoot level resulting in higher dry weight; Enhanced chlorophyll, carotenoids, and proteins content; Improved enzyme activity mainly GOGAT, CS, and PAL.	[97]
*Chlorella* *vulgaris; * *Scenedesmus quadricauda*	Microalgal extract used as seed soaking solution in Petri dishes	Sugar beet seeds (*Beta vulgaris* subsp. vulgaris)	Increased germination indices as well as mean germination time; Enhancement of root volume level and diameter.	[53]
*Chlorella ellipsoidea*	Soil drench with an acid hydrolysis extract	Tomato (*Solanum lycopersicum*)	70.88% and 29.11% increases in root length and dry weight; 53.6% increase in shoot length; 40.36% increase in chlorophyll a leaf content; Enhanced root concentration of nitrogen, phosphorus, and potassium.	[59]
*Chlorella * *vulgaris; * *Tetradesmus dimorphus*	Foliar application of microalgal suspensions in amended soils	Common bean (*Phaseolus vulgaris*)	Plant height increased by 29.6% and dry weight by 37.28%; Improved total carbohydrate and protein content; Increased pod number per plant, seed number per pod, and pod dry weight.	[50]
*Desmodesmus subspicatus*	Foliar application of aqueous extract	Tomato (*Solanum lycopersicum*)	Improved total root length and fresh and dry root biomass ratio, Increased foliar area; Enhancement of hypocotyl length and volume.	[98]
*Chlorella *sp. (MACC-360 and MACC-38)*;* *Chlamydomonas reinhardtii* (cc124)	Soil drench with live microalgae cells	Barrelclover (*Medicago truncatula*)	Higher number of flowers and leaf size; Increased plants’ fresh weight; Enhanced shoot length and pigment content; *Chlorella* application resulted in higher biostimulant effects compared to *C. reinhardtii*.	[48]
*Chlorella vulgaris*	Foliar spray with microalgal extract	Lettuce (*Lactuca sativa*)	Enhanced shoot parameters such as fresh and dry weights, pigment, ashes and protein content; Enhanced root parameters such as dry matter, protein and ashes. Overall improvement of primary and secondary metabolisms.	[52]
*Chlorella vulgaris*	Foliar spray and soil drench with microalgal extract (with/without cowdung)	Tomato (*Solanum lycopersicum*)	Significant increases in fruit quality (fruit length, fruit diameter, number of seed/fruit, and seeds weight/fruit); Increases in total soluble sugars, L-ascorbic acid, and total protein content of fruits; Enhanced mineral content and improved shelf life.	[57]
*Chlorella vulgaris*	Foliar spray and soil drench with microalgal extract	Lettuce (*Lactuca sativa*)	Both methods successfully increased growth parameters such as shoot height, number of leaves, and root length; Pigment and protein contents were also increased; Enhanced enzyme activity and improved metabolic activity.	[99]
*Chlorella vulgaris*	Foliar spray with algal cell liquid extract	Green gram (*Vigna radiata* L.)	Shoot and root length were increased; Enhanced plant branches, plant leaves, and leaf area index; Increased fresh weight of root nodules; Overall improvement of leaf chemical composition (N, K, P, Indole, Phenol); Enhancement of plants water absorption index, solubility index, plus water and oil absorption.	[100]
*Chlorella vulgaris*	Foliar spray with *Chlorella* suspension (CS), *Chlorella* biomass (CB), and *Chlorella*-free supernatant (CFS)	“Red Russian” Kale (*Brassica napus* var. Pabularia)	Increased fresh and dry weights (CB); Higher number of leaves (CFS); Increased total carotenoid content (CB and CS), chlorophyll content (CFS), flavonoid and total phenolic content (CFS).	[101]

**Table 5 plants-13-00159-t005:** Green microalgae-based biostimulants as stress alleviators in different crops.

Green Microalgae Strain	Stress	Crop Plant	Application Method	Biostimulant Effects	Reference
*Dunaliella salina*	Salt stress	Tomato (*Solanum lycopersicum*)	Foliar spray with polysaccharides extract	▪ Mitigation of salt stress by increasing shoot and root systems; ▪ Improved chlorophyll a content in stressed plants; ▪ Mitigation of salt stress by increasing proline content; ▪ Decreased phenolic content and improved enzymatic activity by increasing CAT, SOD, and POD.	[84]
*Chlorella * *vulgaris*	Drought stress	Broccoli (*Brassica oleracea*)	Foliar spray with microalgal extract	▪ Enhancement pigment content and total carotenoids; ▪ Decreased membrane damage by lowering malondialdehyde levels; ▪ Alleviation of oxidative stress by increasing enzyme activity.	[102]
*Desmodesmus* sp.; *Dunaliella salina*	Biotic stress	Tomato (*Solanum lycopersicum*)	Injection of microalgal polysaccharides extract	▪ Improved protein content in stressed plants by 55.01% when using *D. salina* extract; ▪ *Desmodesmus* sp. and *D. Salina* 067 extracts increased chitinase activity by 19.95% and 18.63%; ▪ Increased 1,3-β, glucanase activity with *D. salina* extracts.	[103]
*Chlorella * *sorokiniana; Chlamydomonas reinhardtii*	Nitrogen deficit and drought stress	Maize (*Zea mays*)	Seedling soaked in a nutrient solution supplemented with algae freeze-dried biomass	▪ Treatment with C. reinhardtii powder improved Mn^2+^ content in shoots, thus improving ROS scavenging activity; ▪ *C. sorokiniana* treatment increased the number of secondary roots as well as their length in low-nitrogen conditions; ▪ Higher Mn^2+^ and Cu^2+^ accumulation in shoots and roots under low-nitrogen conditions in plants treated with freeze-dried *C. sorokiniana*.	[104]

**Table 6 plants-13-00159-t006:** Diatom-based biostimulant/biofertilizers as growth promoters and stress alleviators in different crops.

Diatom Strain	Application Method	Crop Plant/Seeds	Biostimulant Effects	Reference
*Phaeodactylum* *tricornutum*	Seeds soaking with polysaccharides extract	Bell pepper (*Capsicum annuum* L.)	A 41% increment in mean germination time in comparison to control; Under salinity stress, superoxide radicals and lipid peroxidation decreased significantly; Increased SOD, CAT, and GPx activity increased under high-salinity conditions.	[106]
Diatom (Unspecified species)	Foliar spray with Diatoms suspension	Washington navel orange (*Citrus sinensis*) and Murcott Tangor (*C. reticulata x sinensis*) transplants	Increased agronomic features such as plant height, number of leaves, and leaf area; Higher chlorophyll and carotenoid content; Enhanced carbohydrate content in shoots and leaves.	[107]
*Navicula* sp.	Direct watering, spraying, and watering + spraying with sonicated extract of *Navicula* sp.	Willow (*Salix viminalis*); Jerusalem artichoke (*Helianthus * *tuberosus*); Virginia mallow (*Sida * *hermaphrodita*)	A 12–25% increment in plant height with tree variants of treatment on all plants; Higher chlorophyll content as well as a higher chlorophyll content index (CCI) in leaves; Increased activity of phosphorus regulation enzymes.	[108]

**Table 7 plants-13-00159-t007:** Examples of microalgae- and cyanobacteria-based consortia and combinations used as plant biostimulants.

Consortia/Combination	Crop Plant	Biostimulant Effects	Reference
**Microalgae and cyanobacteria:** *Chlorella* sp. + *Scenedesmus* sp. + *Spirulina* sp. + *Synechocystis* sp.	Tomato (*Solanum* *lycopersicum*)	Faster germination; Increased shoot and root length, fresh and dry weight in seeds primed with 40% concentration; Foliar spraying with 60% concentration enhanced plant total height, root length as well as chlorophyll content; Increased moist content, phosphorus content, potassium content, and sodium content.	[51]
**Cyanobacteria and diazotrophic bacteria co-inoculation:** *Anabaena cylindrica +* *Rhizobium freirei* *+ Rhizobium tropici +* *Azospirillum brasilense*	Common bean (*Phaseolus* *vulgaris*)	Increased plant height, shoot dry matter, and root length and volume; Improved nitrogen accumulation in shoots; Higher number of nodules at flowering, number of grains per pod, and weight of hundred seeds. Increased yield (62–84%) with tri-inoculation treatment.	[118]
*Anabaena cylindrica + Azospirillum brasilense*	Maize (*Zea mays*)	Increased mass of 1000 grains and average number of grains per row of ear of maize; Leaf content of phosphorus and nitrogen as well as chlorophyll content enhanced under the co-inoculation.	[119]
Cyanobacteria SAB-B866 *(Nostocaceae* Family*) +* *Pseudomonas putida-BIO175 +* *Pantoea cypripedii- BIO175*	Tomato (*Solanum lycopersicum* San Pedro variety)	Greater aerial development in treated plants with both inoculums; Treatment with cyanobacteria/*P. cypripedii* and cyanobacteria/*P. cypripedii* increased fresh weight, leaf number, and stem diameter.	[120]
**Microalgae and substances combination:** *Chlorella* sp. and Vermicompost combination	Maize (*Zea mays*)	Enhanced soil aggregate stability with combination of vermicompost and *Chlorella* sp.; Increased organic carbon content in soil; Enhanced soil fertility.	[121]
*Scenedesmus subspicatus* and humic acids combination	Mung bean (*Vigna radiata*); Onion (*Allium cepa* L.)	A 39% increment in root length in mung bean plants; Increased onion growth parameters under pot conditions; Enhanced onion bulb caliber, sugars and protein content under field conditions.	[122]

**Table 8 plants-13-00159-t008:** Examples of some commercially available microalgae- and cyanobacteria-based biostimulant products and their characteristics.

Product	Microalgae	Composition	Application Method	Biostimulant Effects	Country
AGRIALGAE^®^ Premium Rooting	Microalgae combination	▪ Phytohormones; ▪ Vitamins; ▪ Minerals; ▪ Peptides; ▪ Polyunsaturated fatty acids; ▪ Polysaccharides.	Soil application	▪ Root formation promotion; ▪ Increased nutrient uptake efficiency; ▪ Rhizosphere enrichment.	Spain
AGRIALGAE^®^ Premium Sprouting	Microalgae combination	▪ Phytohormones; ▪ Vitamins; ▪ Minerals; ▪ Peptides; ▪ Polyunsaturated fatty acids; ▪ Polysaccharides; ▪ Ca; Mg; Fe; Mn.	Foliar application	▪ Sprouting improvement; ▪ Photosynthetic capacity increasement.	Spain
AGRIALGAE^®^ Premium Flowering	Microalgae combination	▪ Phytohormones; ▪ Vitamins; ▪ Minerals; ▪ Peptides; ▪ Polyunsaturated fatty acids; ▪ Polysaccharides; ▪ B; Zn.	Foliar and soil application	▪ Flower induction optimization; ▪ Improvement of pollination rate; ▪ Flowering homogenization.	Spain
AGRIALGAE^®^ Premium Fruit Setting	Microalgae combination	▪ Phytohormones; ▪ Vitamins; ▪ Minerals; ▪ Peptides; ▪ Polyunsaturated fatty acids; ▪ Polysaccharides; ▪ B.	Foliar and soil application	▪ Increase fruit setting rate; ▪ Pollen fertility; ▪ Reduced premature fall of fruits.	Spain

## Data Availability

All research data are included in the manuscript.

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
