# Peer review of "A Comprehensive Review of Microalgae and Cyanobacteria-Based Biostimulants for Agriculture Uses"

_plants, 2024, doi:10.3390/plants13020159_

Round 1

Reviewer 1 Report

Dear Authors,

I believe the article is well-written; it accurately captures and adequately analyzes the advancements and drawbacks in this field. I consider this paper to be a step forward in knowledge due to the ideas it introduces. However, before publication acceptance, there are some areas that need improvement. One of them is the consideration that cyanobacteria are not strictly classified as microalgae. The authors should rewrite the title and certain parts of the article to correct this. Additionally, greater emphasis should have been placed on the consortium of microalgae with nitrogen-fixing organisms, as it has recently been found to hold significant biotechnological potential.

- In the paper I received, the lines were not numbered, which makes the reviewing process quite challenging.

Majors:

-“ Cyanobacteria and eukaryotic microalgae are part of a highly diverse polyphyletic group of unicellular and multicellular photosynthetic microorganisms” I don't understand this sentence. Aren't microalgae, by definition, single-celled?

- I believe that in the introduction, there should be a clearer definition of what a biostimulant is and how it differs from a biofertilizer.

-Ín the section: 2.1. Selection of microalgae strains with high biostimulant potential:

I miss in this section that there is not a deeper discussion about nitrogen as an important factor when selecting the chosen strains of microalgae. Nitrogen fixation in cyanobacteria is mentioned; recently, a publication addressing the consortium between microalgae and nitrogen fixing organisms and its applications, which covers this topic, has been released. DOI10.3390/plants12132476

-In the section: 2.2. Microalgae cultivation and biomass production:

- “Microalgae can be cultivated under different metabolic growth models depending on energy and carbon sources, including autotrophy, heterotrophy, mixotrophy, and photoheterotrophy [18]. Following are the three main production systems used in microalgae cultivation: heterotrophic, photo-autotrophic, and mixotrophic [19”]. I think these two sentences are redundant. What does "photoheterotrophically" mean? Isn't it the same as mixotrophic? Please clarify.

-Comment on the issues of microbial contamination that open ponds face.

-“Exorbitant costs of biomass production” Mention that one of these costs is the nitrogen source.

-Table 1: "supplemented with centrate” What is centrate?

-“environmental conditions, culture conditions especially energy and carbon supply are all” change to environmental conditions, culture conditions especially energy,  carbon  and nitrogen supply are all.

-The caption of Figure 1 needs an explanation of what the authors intend to convey with the depiction of the Eppendorf tube and its contents. What does this content represent?

- Although I am not a native English speaker, I can see that the article is well-written. However, some sentences appear to be overly complex. I suggest to the authors that they have the article reviewed by a native English speaker. An example of this are the following sentences.

-“Over and above the cultivation of microalgal biomass, pretreatments and extraction, another hurdle in the process line of microalgae-based biostimulant formulation, which is oftentimes overlooked, is the storage effect and the shelf life of these biostimulants” changed to “In addition to the cultivation of microalgal biomass, pretreatments, and extraction, another often overlooked challenge in the process of formulating microalgae-based biostimulants is the storage and shelf life of these products”.Principio del formulario

- “In light of this result, it is elementary to incite local efforts to vulgarize the use of microalgae in the cropping system especially in countries where agriculture is an economic pillar”. Change to: In light of this result, it is essential to encourage local efforts to promote the use of microalgae in the cropping system, especially in countries where agriculture is an economic pillar.

-“ In terms of use of microalgae as biostimulants, cyanobacteria and green alga are the most used, while the use of diatoms is still in its early stages” I believe that cyanobacteria are not strictly considered microalga. I believe and advise that it would be better to change the title for that reason.  “A comprehensive review of microalgae and cyanobacterial-based biostimulants for agriculture uses”. In relation to this"Microalgae-based biostimulants (MBs)" - this abbreviation is only used in the abstract. I suggest it be better defined as "Microalgae and Cyanobacteria-based biostimulants (MCBs)" and used throughout the manuscript to make it clear that cyanobacteria are also included.

- In this section: 3.1. Cyanobacteria use as plant biostimulants: It would be convenient to mention that recently, mechanisms of cooperation between microalgae and nitrogen-fixing organisms like cyanobacteria have been described, along with their potential biotechnological applications.

- “Algal and microbial consortia are well known for increasing plants’ capacity to absorb water and nutrients as well as being involved in carbon Exchange” change to Algal and microbial consortia are well known for increasing plants’ capacity to ab-sorb water and nutrients as well as being involved in carbon and nitrogen exchange.

- in the section “Use of microalgae-based consortia/associations as plant biostimulants” Comment that microorganisms that enhance the growth of microalgae have been described. Specify that these microorganisms that promote the growth of microalgae are called Microalgae Growth Promoting Microorganisms (MGPM). Provide a more in-depth explanation of the role of MGPM in microalgae cultivation.

- Table 7 is somewhat confusing, as it has three different headings. I believe that Synechocystis sp. is considered a cyanobacterium and not a microalga. In Table 3, I don't understand the significance of the association of Chlorella sp. and Scenedesmus subspicatus with Vermicompost and Humic acids, respectively. Could you please clarify this? As I understand, the table is about consortia, so why the humic acids?. Anabaena cylindrica, I believe, is a cyanobacterium.

Minors:

- In Figure 1, I suggest changing the color of the algae biomass tube from red to green

-Botryoccoccus sudeticus, italic please.

-“Soxhlet extraction” please defined what is this

-“Ergo the biostimulant effects” please change to: Hence the biostimulant effects.

Moderate editing of English language required

Reviewer 2 Report

In this manuscript authors show that the research progress and application status of microalgae-based biostimulants, comprehensively covering all aspects from production to industrialization. Generally, the topic is interesting, but more comprehensive discussion may be necessary to complete the review.

Major concerns:

1. In 2.2, the last paragraph indicates that biorefinery approach have been incorporated in agriculture, is there a current example of application on field crops, such as rice, maize, etc?

2. In the third part, authors status the role of various microalgae-based biostimulants in agriculture, it is curious what are the pros and cons of these different microalgae-based biostimulants? What factors should be considered when determining the type of microalgae in specific applications?

3. In the fifth part, this manuscript states that “a survey about consumers attitude toward microalgae based agricultural products highlighted that farmers in the region of Almeria (Spain) and Livorno (Italy) were not in favor of building microalgal production facilities in their properties, however the use toward microalgae based agricultural products such as biostimulants, biofertilizers and aquafeed in both locations was acceptable especially in Livorno (Italy).  Does it mean that people believe that establishing microalgae production factories will cause pollution?

4. In the seventh part, authors summarize the bottlenecks and use limitations of microalgae biostimulants, this section should be supplemented appropriately.

5. Proper modification of English expression.

Proper modification of English expression will be necessary.

Reviewer 3 Report

I recommend that this paper be accepted after minor revision.

Some editing for English language is required throughout the manuscript due to too some mistakes. Please, pay attention.

Line 3: there is a dot. Why?

Line 52: algal extract…why the three dots? There is some reason? The same error on line 478?

On Paragraph 6. Regulations and legal framework of biostimulants for agriculture uses could be better a more detailed explanation of the efficacy of the biostimulant under determined conditions. There is only one reference regarding this aspect.

Some editing for English language is required throughout the manuscript due to too some mistakes. Please, pay attention.

Round 2

Reviewer 1 Report

Dear Authors,

I consider that the authors have responded correctly to all of my suggestions and comments. There is only a small mistake: reference 76 refers to the preprints, and the peer-reviewed version is now available: https://doi.org/10.3390/plants12132476.